# Few-shot Learning for Cardiac Segmentation via Self-supervised Multi-task Learning

**Jiahang Xu**          JHXU18@FUDAN.EDU.CN  and  **Xiahai Zhuang**          ZXH@FUDAN.EDU.CN
*School of Data Science, Fudan University, Shanghai, China*

## Abstract

In this paper, we propose two novel self-supervised tasks, which encode the spatial and modality appearance information of cardiac images, respectively. Furthermore, we propose to ensemble multiple self-supervised tasks in a multi-task learning framework to learn more effective semantic representation for cardiac segmentation. The proposed approach was validated on the late gadolinium enhancement (LGE) cardiac magnetic resonance (CMR) images from the public MICCAI 2019 MSCMRseg Dataset, and was compared with two popular self-supervised tasks, including context inpainting (CI) and context restoration (CR). Results demonstrated that the proposed self-supervised tasks, as well as hybridized multi-task learning strategies, are effective in few-shot cardiac segmentation.

**Keywords:** Self-supervised Learning, Multi-task Learning, Multi-modality CMR, Cardiac Segmentation

## 1. Introduction

Medical image segmentation is one of the most highlighted processes in medical image analysis. However, manual annotation is difficult even for the experts. To handle the segmentation with limited labeled images and plenty of unlabeled images, *self-supervised learning* (SSL) is proposed, i.e. learning image representation from the supervision tasks generated from data themselves. SSL provides a proxy loss, by which the network is forced to learn the semantic representations of the data. To this end, several SSL tasks have been designed for deep learning, such as context inpainting (CI) (Pathak et al., 2016) and context restoration (CR) (Chen et al., 2019). In these two tasks, fragments of the images go through a disturbance, and are forced to be restored by the target network.

Generally, SSL handles each image independently. However, slices in a 3D medical images or with different modalities usually share similar anatomical and pathological structures. Therefore, encoding spatial information and modality-related appearance information could help better "understand" the images. Furthermore, one could combine multiple different SSL tasks and share their representations via a *multi-task learning* (MTL) framework.

In this work, we proposed two novel SSL tasks for few-shot cardiac segmentation, i.e. spatial encoding (SE) to encode spatial information, and modality encoding (ME) for modality appearance information. Then we hybridized these SSL tasks via the framework of MTL to learn more comprehensive features from unlabeled images. Finally, we verified the performance of the proposed method on the cardiac segmentation task using the late gadolinium enhancement (LGE) cardiac magnetic resonance (CMR) images. The results demonstrate great potential of the proposed methods with small data volumes.

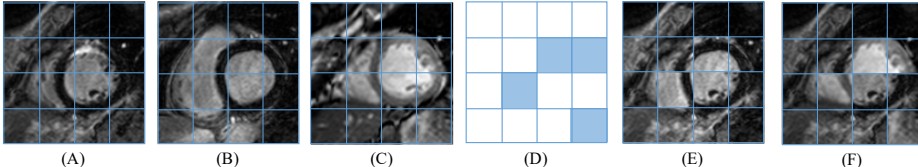

Figure 1: Implementation of SE and ME tasks. (A): a middle slice (the *major* slice); (B): a basal slice (the *replacement* slice for SE); (C): corresponding bSSFP slice of (A) (the *replacement* slice for ME); (D): replaced patches (marked as blue); (E): resulting SE image with (A) and (B); (F): resulting ME image with (A) and (C).

## 2. Methodology

In SSL, networks are encouraged to capture the image context. Here, we introduce the spatial encoding (SE) and modality encoding (ME) tasks, which are illustrated in Fig 1. In the two tasks, several patches from the *major* slice are randomly selected and replaced by the corresponding patches from the *replacement* slice.

**SE task:** SE task replaces patches from a *major* slice with the *corresponding* patches of a different slice from the same volume, and then restore the *major* slice by network.

**ME task:** ME task replaces patches of a *major* slice with the *corresponding* patches from different modalities of the same subject, and then restore the *major* slice by network.

The SE and ME task could help verify the current spatial position and extract the cross-modality representation, respectively. Furthermore, we ensemble CI, CR, SE, and ME in a common MTL framework to enhance the representation learning.

After training the SSL tasks, the learned representation is deployed to the downstream segmentation task. We applied a standard U-net architecture (Ronneberger et al., 2015) with different output layers for the pre-training and the segmentation phases. We used the MSE loss in the pre-training phase, and used a combination of the Dice loss and the binary cross-entropy (BCE) loss in segmentation phases.

## 3. Experiment and Results

We evaluated the proposed method on the MICCAI 2019 Multi-sequence Cardiac MR Segmentation (MSCMRseg) Dataset[1] (Zhuang, 2016, 2018). We randomly split the LGE CMR images into sets of 5/5/30 subjects as training/validation/test set. Furthermore, 150 additional unlabeled balanced Steady-State Free Precession (bSSFP) CMR images from the MICCAI 2017 Automated Cardiac Diagnosis Challenge (ACDC) Dataset[2] (Bernard et al., 2018) were used for SSL training. The code was implemented using PyTorch[3].

Table 1 presents the Dice coefficients, for the myocardium (Myo), left ventricle (LV), and right ventricle (RV), with the different SSL strategies. Note that the proposed SE and ME tasks achieved better Dice values than CI and CR, suggesting the spatial information and modality appearance information are useful to the cardiac segmentation tasks. Also,

---

1. https://zmiclab.github.io/mscmrseg19/

2. http://acdc.creatis.insa-lyon.fr/description/index.html

3. pytorch.org

Table 1: Comparisons in 5-shot learning for LGE segmentation with different pre-training strategies. Bold font indicates the best results in a column.

| Method | Dice | | |
|---|---|---|---|
| | Myo | LV | RV |
| w/o Pre-train | $0.703 \pm 0.140$ | $0.848 \pm 0.135$ | $0.716 \pm 0.199$ |
| CI | $0.752 \pm 0.096$ | $0.899 \pm 0.048$ | $0.739 \pm 0.157$ |
| CR | $0.740 \pm 0.109$ | $0.873 \pm 0.086$ | $0.750 \pm 0.166$ |
| SE (Ours) | $0.757 \pm 0.102$ | $0.894 \pm 0.077$ | $0.779 \pm 0.160$ |
| ME (Ours) | $0.730 \pm 0.113$ | $0.866 \pm 0.108$ | $0.763 \pm 0.153$ |
| MTL (Ours) | $\mathbf{0.764 \pm 0.096}$ | $\mathbf{0.909 \pm 0.047}$ | $\mathbf{0.780 \pm 0.153}$ |

MTL performed better than all the single SSL task, indicating the hybridization of multiple tasks encouraged the network to learn representation in a more comprehensive manner.

## 4. Conclusion

In this work, we proposed two self-supervised tasks, i.e. spatial encoding and modality encoding. Then, we integrated the multiple SSL tasks by MTL strategy, based on which we implemented cardiac segmentation on LGE images. Results showed the evident effectiveness of the proposed tasks.

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
