# OpenReview forum: "Few-shot Learning for Cardiac Segmentation via Self-supervised Multi-task Learning"
_MIDL.io/2021/Conference/Short — Submitted to MIDL 2021_

### Official Review · Reviewer_Q8TA · 2021-04-26

**Confidence:** 3
**Final Rating:** 2

**Summary:**

The authors propose the use of two self-supervision tasks to improve the performance of few-shot cardiac MR segmentation. Both tasks replace 2D patches of image slices - the “modality encoding” task replaces these by patches of a different modality while the "spatial encoding" task replaces them with patches from a different slice in the same volume - and the model should learn to reconstruct the original images. The tasks are compared to two other self-supervised tasks: context inpainting and context restoration. In addition, a method variation combining the four self-supervised losses is evaluated and found to outperform models trained with only one proxy task. The models are trained with 5 annotated subjects, and the self-supervision loss is additionally trained with 150 subjects from a different dataset.

**Strengths:**

- The paper explores few-shot learning, which is a relevant topic for the medical image community.
- The “modality encoding” task is a reasonable strategy to take when multi-modality data is present.
- Figure 1 greatly assists the reader in understanding both proposed proxy tasks.


**Weaknesses:**

- The main weakness of the paper is that the proposed SE task is very similar to jigsaw puzzle solving (see Noroozi, M., & Favaro, P. (2016, October). Unsupervised learning of visual representations by solving jigsaw puzzles. In European conference on computer vision (pp. 69-84). Springer, Cham.). Yet the authors do no refer to this existing approach and instead introduce SE as a novel idea.
- The authors mention that “the proposed SE and ME tasks achieved better Dice values than CI and CR”. Yet this is only true for right ventricle segmentation, and partially for myocardium segmentation.
- It is doubtful whether comparing the proposed tasks to only context inpainting and context restoration is sufficient, and neither of these tasks encourage the model to learn geometric information.


**Deanonymize Review:**

no

**Detailed Comments:**

- The first paragraph of the introduction could be significantly shortened.
- A clear reference to the jigsaw puzzle solving method should be included, and the authors should clearly state why their proposed method is better suited to cardiac MR segmentation.
- It is unclear how context inpainting is implemented for the data at hand. I would suggest including one or two sentences to explain this in detail.


**Justification Of The Rating:**

The authors suggest that the novelty of the proposed self-supervision tasks is a main contribution, yet similar methods exist for which there are not adequate references. In addition, the two proxy tasks that are compared against the proposed tasks are context inpainting and context restoration, which do not encourage the model to learn geometric information and are therefore not sufficient to validate the method.

**Paper Type:**

both

**Special Issue:**

no

---

### Meta-Review · Area_Chair_rvyo · 2021-05-11

**Recommendation:** Reject
**Confidence:** 4

**Metareview:**

The underlying idea of combining a jigsaw solving pretext task with cross-modal multi-task learning is attractive. The reviewer, however, highlights important issues e.g. a missing reference to the jigsaw paper and when taking this into account very limited gain of the cross-modal task. This may be due to imperfect alignment, but moreover the choice of baseline methods is not state-of-the-art: what about Doersch et al. and following or colorisation with different MRI contrasts? In summary I agree with the reviewer that the paper cannot be accepted in the current form.

---

### Decision · Program_Chairs · 2021-05-11

Reject